# The HIV-1 *vpr* R77Q Mutant Induces Apoptosis, G_2_ Cell Cycle Arrest, and Lower Production of Pro-Inflammatory Cytokines in Human CD4+ T Cells

**DOI:** 10.3390/v16101642

**Published:** 2024-10-21

**Authors:** Antonio Solis-Leal, Dalton C. Karlinsey, Sidney T. Sithole, Jack Brandon Lopez, Amanda Carlson, Vicente Planelles, Brian D. Poole, Bradford K. Berges

**Affiliations:** 1Department of Microbiology and Molecular Biology, Brigham Young University, Provo, UT 84602, USA; a.solisleal@gmail.com (A.S.-L.); dckarlinsey@gmail.com (D.C.K.); sydyten@byu.edu (S.T.S.); jackblc@student.byu.edu (J.B.L.); amandac1302@gmail.com (A.C.); brian_poole@byu.edu (B.D.P.); 2Department of Pathology, Division of Microbiology and Immunology, University of Utah School of Medicine, Salt Lake City, UT, USA; vicente.planelles@path.utah.edu

**Keywords:** HIV, AIDS, *vpr*, apoptosis, necrosis, inflammation, immune activation

## Abstract

Acquired immunodeficiency syndrome (AIDS) occurs when HIV depletes CD4+ helper T cells. Some patients develop AIDS slowly or not at all, and are termed long-term non-progressors (LTNP), and while mutations in the HIV-1 Viral Protein R (*vpr*) gene such as R77Q are associated with LTNP, mechanisms for this correlation are unclear. This study examines the induction of apoptosis, cell cycle arrest, and pro-inflammatory cytokine release in the HUT78 T cell line following infection with replication-competent wild-type strain NL4-3, the R77Q mutant, or a *vpr* Null mutant. Our results show a significant enhancement of apoptosis and G_2_ cell cycle arrest in HUT78 cells infected with R77Q, but not with WT NL4-3 or the *vpr* Null strain. Conversely, HUT78 cells infected with the WT virus show higher levels of necrosis. We also detected lower TNF and IL-6 release after infection with R77Q vs. WT. The apoptotic phenotype was also seen in the CEM cell line and in primary CD4+ T cells. Protein expression of the R77Q *vpr* variant was low compared to WT *vpr*, but expression levels alone cannot explain these phenotypes because the Null virus did not show apoptosis or G_2_ arrest. These results suggest that R77Q triggers a non-inflammatory apoptotic pathway that attenuates inflammation, possibly contributing to LTNP.

## 1. Introduction

The *vpr* gene is found in many types of lentiviruses, including Human Immunodeficiency Virus type 1 (HIV-1), HIV-2 and Simian Immunodeficiency Virus. It has a critical role in pathogenesis and replication, and is classified as a viral regulatory protein [1,2]. The *vpr* gene has been shown to be highly conserved across many types of lentiviruses, suggesting the biological importance of *vpr* in the viral life cycle [3,4].

Although *vpr* is a late gene, the mature virion carries many copies of the protein inside the capsid, and thus it is present/active during early stages of infection [5,6]. *vpr* carries out different functions during the viral infections cycle. It is part of the pre-integration complex, which facilitates the integration of viral DNA into chromatin [7]. It also likely plays a crucial role in counteracting cellular antiviral factors; this activity is highly related to its morphology and interactions with other viral proteins or host factors [8]. Furthermore, the presence of *Vpr* has been shown to enhance virus replication in T cells [9]. This effect could stem from several of the interactions that *vpr* has with the host cell, including *vpr*-mediated G_2_ phase cell cycle arrest [10,11,12,13] or through transcriptional modulation [14,15,16,17]. During the G_2_ phase, the HIV long terminal repeat promoter is more active [18], suggesting that viral-induced G_2_ arrest could be a tool to enhance viral replication.

One of the other functions most commonly associated with *vpr* is the regulation of apoptosis in infected cells [19,20]. Early studies showed that *vpr* can cause the permeabilization of the permeability pore complex (PTPC), resulting in the release of apoptotic factors from the mitochondrial membrane [20,21]. However, it was later shown that *vpr*-induced depolarization of the mitochondrial membrane was not a result of *vpr* binding to PTPC, but rather due to activation of the Bax protein, which is a result of stress signals sent by the Ataxia Telangiectasia and Rad3-related (ATR) DNA damage response pathway [22]. The ATR pathway responds to DNA damage, and when activated, it initiates a signaling cascade that stalls the cell at a cycle checkpoint [23]. Activated ATR has also been shown to phosphorylate Breast Cancer type 1 susceptibility protein (BRCA1), and this in turn leads to the upregulation of Growth Arrest and DNA Damage-inducible alpha protein (GADD45α), which eventually leads to Bax activation and the release of apoptotic factors from the mitochondria. Knocking down ATR expression with RNAi abrogated both the G_2_ arrest and apoptosis associated with *vpr* expression, showing that the processes happen concomitantly [22,23,24]. It has been hypothesized that ATR is activated in response to *vpr*-induced herniations in the nuclear membrane [24,25]. Despite both processes being regulated by ATR activation, the *vpr* induction of G_2_ arrest and apoptosis are independent processes [18,26,27,28]. It has been observed that HIV-infected cells are more commonly apoptotic during G1 or S phase, while cell cycle arrest occurs in the M or G_2_ phase [21,27,29,30]. Both virion-associated and de novo-synthesized *vpr* are able to induce apoptosis and cell cycle arrest [27,31]. Additionally, secreted *vpr* has been suggested to cause apoptosis by the permeabilization of cellular membranes to calcium and magnesium [32,33,34]. *vpr*- and reverse transcription-induced apoptosis have been shown to be the main causes of cell death in peripheral blood resting CD4+ T cells, potentially contributing to the bystander cell death associated with HIV infections [35].

Acquired immunodeficiency syndrome (AIDS) is a consequence of the tropism of HIV, which targets CD4+ T cells. These cells are essential for coordinating adaptive immune responses to pathogens, but HIV infection can cause a decrease in their concentration in blood to less than 200 CD4+ T cells/μL of blood, compared to about 1000 CD4+ T cells/μL of blood in healthy individuals [36]. People living with HIV (PLWH) that maintain normal levels of CD4+ T cells for many years are classified as long term non-progressors (LTNP) [37]. Additionally, LTNPs have been reported to have low chronic immune activation, which is the hallmark of HIV-1 progression despite the high HIV-1 viral load, suggesting that chronic immune activation is a more superior prognostic indicator of HIV-1 progression [38]. Mutations in *vpr* have been proposed as a causative agent of the phenotypic differences observed in PLWH [39,40], but there is much that is not understood due to the complexity of both viral and host genetics. Various mechanisms have been proposed to explain how *vpr* could enhance disease progression: promoting early T cell activation by enhancing the Nuclear Factor of Activated T cells (NFAT) to prime non-activated T cells for productive infection [41]; inhibiting T cell proliferation and enhancing cell death [3]; and inducing the infection and activation of virus production from latency in macrophages, creating drug-resistant HIV reservoirs [42,43,44].

The *vpr* R77Q mutant has been correlated with LTNP. This mutation has been found in several LTNPs and has also been reported to show a reduction in T cell depletion [9,39,45,46]. Other studies, however, have shown that this mutation does not delay AIDS progression in subjects receiving antiretroviral therapies [47,48], increasing the uncertainty around the cytopathogenicity of this mutation.

In this study, we analyzed the effect of the *vpr* R77Q mutation on HIV-1-infected human CD4+ T cells for several pathogenic phenotypes in terms of the rate of viral replication, induction of apoptosis, induction of G_2_ cell cycle arrest, and release of pro-inflammatory cytokines to better understand the role that *vpr* mutants play in the induction of cell death and pro-inflammatory pathways.

## 2. Materials and Methods

### 2.1. Plasmids

HIV-1 pNL4-3 was obtained through the NIH AIDS Reagent Program, Division of AIDS, NIAID, NIH: the wild-type (WT) virus used was the HIV-1 NL4-3 Infectious Molecular Clone (pNL4-3) from Dr. Malcolm Martin (Cat# 114) [49,50,51]. The HIV-1 NL4-3 R77Q (CAA codon; WT has CGA) *vpr* mutants were a gift from Dr. Velpandi Ayyavoo (University of Pittsburgh). The region surrounding the *vpr* gene was cloned into the pUC19 vector (Addgene #50005, Watertown, MA, USA) using the EcoRI and SbfI sites (New England Biolabs, Ipswich, MA, USA) for site-directed mutagenesis (SDM). SDM was accomplished using back-to-back primers with single point mismatches to change the ATG codon to GTG for the Null mutant in order to not affect any overlapping genes in the viral genome (Forward: CAGAGGACAGGTGGAACAAGC; Reverse: TCAGTTTCCTAACACTAGGC). After SDM, the *vpr* gene was removed from pUC19 using the same enzymes and ligated back into its original frame in the pNL4-3 vector. The desired mutation and in-frame ligations were confirmed by Sanger sequencing. One shot Stbl3 *E. coli* cells (Invitrogen, Carlsbad, CA, USA, C737303) were used for the transformations to avoid plasmid recombination.

### 2.2. Cell Culture

HEK293FT cells were maintained at 37 °C, 5% CO_2_ in DMEM (Sigma-Aldrich, Milwaukee, WI, USA) supplemented with 10% fetal bovine serum (FBS; HyClone, Logan, UT, USA), 2% glutamine, 1% G418 (Teknova, Hollister, CA, USA), and 1% penicillin/streptomycin (P/S). HUT78 and CEM cells were maintained at 37 °C, 5% CO_2_ in RPMI (Mediatech, Austin, TX, UDA) supplemented with 10% FBS, 1% glutamine, and 1% P/S. Ghost R3/X4/R5 cells were maintained at 37 °C, 5% CO_2_ in DMEM, supplemented with 10% FBS, 1% P/S, 500 μg/mL G418, 100 μg/mL hygromycin, and 1 μg/mL puromycin. (Ghost cells were obtained through the NIH AIDS Reagent Program, Division of AIDS, NIAID, NIH: GHOST (3) CCR3+ CXCR4+ CCR5+ Cells from Dr. Vineet N. KewalRamani and Dr. Dan R. Littman (cat# 3943) [52]).

### 2.3. Transfection and Viral Titration

HEK293FT cells were transfected using the calcium phosphate method; the virus was collected 48 h post-transfection. Viral concentration was determined by titration using the Ghost R3/X4/R5 cell line and the methods described in the NIH AIDS Reagent Program (Rockville, MD, USA).

### 2.4. HIV-1 Infections and Flow Cytometric Analysis

Cells were infected with 0.1 MOI (cell cycle analysis) or 0.01 MOI (all other experiments) with polybrene. All samples labeled as “uninfected” herein were treated with the same polybrene concentration as for infected samples. Cells were collected on the specified day of infection for each test and prepared as follows for Annexin V flow cytometric analysis. Cells were washed twice with serum-free PBS, centrifuged for 5 min at 500× *g*, stained with fixable viability dye (FVD) for 30 min at 4 °C, and then washed with FACS staining buffer (FCSB) (140 mM NaCl, 4 mM KCl, 0.75 mM MgCl_2_, 10 mM HEPES, 1% BSA, 0.1% NaN_3_). CaCl_2_ was then added at a concentration of 1.5 mM for 10 min before staining with fluorochrome-conjugated Annexin V for 15 min at 25 °C. Cells were then washed with FCSB and 1.5 mM CaCl_2_, fixed with Solution A and 1.5 mM CaCl_2_ for 30 min, washed with FCSB, and permeabilized with Solution B (Fix & Perm, Nordic MUbio, Susteren, The Netherlands, GAS-002). An Anti-p24 antibody was added at a final dilution of 1:400, and cells were incubated for 15 min at 25 °C. Cells were then washed twice with FCSB, resuspended, and analyzed by flow cytometry using a Beckman Coulter Cytoflex Cytometer (Brea, CA, USA) [53,54]. The results were analyzed using FlowJo software (version 10.6.2). The dyes used were as follows: KC57 anti-HIV-1 core antigen-PE (Beckman coulter CO604667); Live/Dead fixable Far Red Dead Cell Stain Kit (Invitrogen L10120); and Annexin V FITC conjugate (Invitrogen A13199). For cell cycle analysis, we used the FxCycle PI/RNase Staining Solution (Invitrogen F10797) together with KC57 anti-HIV-1 core antigen-FITC (obtained through the NIH AIDS Reagent Program, Division of AIDS, NIAID, NIH: Anti-HIV-1 p24 Monoclonal (KC57)-FITC from NIAID, DAIDS (cat# 13450)). For TUNEL assays, cells were stained using a FragEL^TM^ DNA Fragmentation Kit, Fluorescent-TdT Enzyme (Millipore QIA39, Saint Louis, MO, USA) following the manufacturer’s protocol.

### 2.5. Immunoblotting

HUT78 cells were washed with ice-cold PBS and lysed in RIPA buffer (ThermoFisher, Waltham, MA, USA), 0.1% SDS (ISCBioExpress, Kaysville, UT, USA), and a protease inhibitor cocktail (ThermoFisher). Proteins were quantified using a BSA Protein Assay Kit (ThermoFisher). Equal amounts of protein were boiled with 6X loading buffer (375 mM Tris HCl, 9% SDS, 50% Glycerol, 0.03% Bromophenol blue) for 5 min. Samples were run through a 12% SDS polyacrylamide gel and then transferred to a PVDF membrane (BioRad). The membrane was blocked with PBS containing 5% non-fat milk for 1 h at 25 °C. *vpr* was probed using an anti-HIV-1 Vpr polyclonal (diluted 1:400) (“The following reagent was obtained through the NIH HIV Reagent Program, Division of AIDS, NIAID, NIH: Polyclonal Anti-Human Immunodeficiency Virus type 1 *vpr* Protein, Residues 1 to 50 (antiserum, Rabbit), ARP-11836, contributed by Dr. Jeffrey Kopp”). GAPDH was probed using rabbit anti-GAPDH (1:4000) (Cell Signaling Technology, Danvers, MA, USA). Both antibodies were detected using goat anti-rabbit HRP-conjugated (1:10,000) (Abcam #ab6721) antibodies. The membrane was stripped using a stripping buffer (200 mM Glycine, 1% SDS, and 1% Tween 20) and blocked with the milk-PBS solution described earlier. p24 was probed using rabbit anti-p24 (1:20,000) (The following reagent was obtained through the NIH HIV Reagent Program, Division of AIDS, NIAID, NIH: Polyclonal Anti-Human Immunodeficiency Virus type 1 SF2 p24 (antiserum, Rabbit), ARP-4250, contributed by DAIDS/NIAID; produced by BioMolecular Technologies, London, United Kingdom). The same secondary antibody from above was used (1:10,000). Detection was undertaken using Western Bright ECL (Advansta, San Jose, CA, USA). A quantitative analysis of the chemiluminescent bands on the western blots was performed using ImageJ software (version 1.54g, National Institute of Health, Bethesda, MD, USA) and normalized to a GAPDH loading control. The relative quantification of the *vpr* band was expressed as a ratio to the uninfected control.

### 2.6. Quantification of Viral Replication

Viral supernatants were collected on days 5 and 7, and viral RNA was extracted using a Viral Nucleic Acid Extraction kit II (Scientific FF10616-GB). The extracted RNA was used as template to produce cDNA using a High Capacity cDNA Reverse Transcription Kit (Applied Biosystems, Carlsbad, CA, USA 4368814) and previously described LTR-specific primers [55]. Finally, a 2X Forget-Me-Not Universal Probe Master Mix with ROX (Biotium, Fremont, CA, USA 31044-1) was used to perform a Q-RT-PCR in an Applied Biosystems StepOnePlus^TM^ Real-Time PCR System Thermal Cycling Block, using the primers and protocol described previously by Rouet et al. [55]. The software used to analyze the data was StepOnePlus Software 2.2.3.Ink, and the results were normalized to RNA copies × mL^−1^.

### 2.7. Measurement of Cytokine Production

HUT78 cells were infected with HIV-1 NL4-3 WT, Null, or R77Q with an MOI of 0.01. A mock infection was also prepared, and all infections were undertaken in triplicate. At 5 days post-infection (dpi), cells were centrifuged and supernatants were collected and stored at −80 °C. Secreted cytokines were quantified using a Cytometric Bead Array Flex Human Soluble Protein Kit (BD, Franklin Lakes, NJ, USA 558264). TNF (BD 560112) and IL-6 (BD 558276) were tagged using beads and fluorescent antibodies that were designed for the appropriate kit. Beads were then analyzed by flow cytometry using a Beckman Coulter Cytoflex Cytometer.

### 2.8. Cord Blood Mononuclear Cell (CBMC) Extraction, HIV-1 Infection and Flow Cytometry

CBMCs were extracted from cord blood collected in liquid Citrate Phosphate Dextrose (CPD) anticoagulant using the Ficoll and SepMate tube gradient density separation method. Briefly, cord blood was diluted with an equal volume of 1× PBS and layered on top of the Ficoll in the SepMate tube followed by centrifugation at 1200 rcf for 10 min according to the manufacturer’s instructions. CD4+ CBMC were then isolated using the EasySep human CD4 positive selection kit II from Stemcell Technologies. CD4+ CBMCs were maintained at 37 °C and 5% CO_2_ in RPMI (Mediatech) supplemented with 10% FBS and 1% P/S overnight before infection. Infections were undertaken in triplicate at an MOI of 0.01 for 7 days. In brief, CBMC-derived CD4+ cells were seeded at 250,000 cells per well and infected with either the WT, R77Q, or Null virus in the presence of polybrene. Uninfected cells were also treated with polybrene. Cells were maintained at 37 °C and 5% CO_2_ in RPMI (Mediatech) supplemented with 10% FBS and 1% P/S. After 7 dpi, cells were harvested and prepared for Annexin V/FVD flow cytometry analysis as described above. Flow cytometry data was analyzed using FlowJo software (version 10.6.2).

### 2.9. Statistical Analysis

Biological samples were analyzed in triplicate using a one-tailed Student’s *t*-test with a significance level of *p* ≤ 0.05.

## 3. Results

### 3.1. Construction of vpr Mutants

All virus strains used in this study were based on the wild-type (WT) pNL4-3 molecular clone, which uses CXCR4 as a co-receptor. The R77Q construct was produced by replacing the WT *vpr* gene with mutated *vpr* sequences from expression plasmids obtained from Dr. Velpandi Ayyavoo [9]. We created the *vpr* Null mutant to study the effects of HIV infections without detectable *vpr* expressions but without any mutations in neighboring genes/reading frames. This was accomplished using site-directed mutagenesis (SDM) to mutate the *vpr* start codon (ATG to GTG, or M1V), resulting in a silent mutation in nucleotide 173 of the overlapping *vif* gene (AGA to AGG; both encode arginine). Sanger DNA sequencing was used to confirm the desired mutations and to show that no additional mutations were introduced during the creation of constructs. Maps of the mutations used are found in Figure 1A. All experiments herein used a virus that was prepared via plasmid transfection, so all viral genome sequences should be highly homogeneous at the start of the experiment.

### 3.2. *vpr* R77Q and Null Mutants Have a Similar Replicative Capacity Compared to WT Virus

To examine the replicative capacity of our mutants in a T cell line, HUT78 cells infected at 0.01 MOI were analyzed for viral replication via a Q-RT-PCR of RNA extracted from infected cell supernatants from 5 and 7 dpi. There was no significant difference in the viral loads between the WT, R77Q, and Null viruses at either 5 or 7 dpi (Appendix A). Although there was a slight decrease in the replicative ability of the Null virus at 7 dpi, the difference, when compared to WT and R77Q, was not statistically significant (R77Q vs. Null, *p* = 0.108), (WT vs. Null, *p* = 0.098) (Figure 1B).

### 3.3. R77Q Virus Expresses Less *vpr* than WT Virus, and Null Virus Makes Undetectable *vpr*

To confirm the expression of *vpr* in cells infected with the WT virus and the R77Q mutant and the lack of expression in the Null mutant, we probed for *vpr* via immunoblotting of infected HUT78 cell lysates collected at 7 dpi with MOI 0.01. Both the WT and R77Q virus produced detectable *vpr*, and no *vpr* was detected in uninfected cells or cells infected with the Null mutant (Figure 2A). *vpr* expression from the R77Q mutant was consistently found at reduced levels compared to the WT virus, across several experiments (see Figure 2A,B and Appendix A). The GAPDH loading control showed that approximately equal amounts of total protein were added for each sample, and the p24 loading control indicated that similar amounts of total HIV-1 proteins were also present in each sample (Figure 2B). A quantitative analysis of the relative *vpr* bands was performed using ImageJ software and showed that there was a significant ~6-fold reduction in the expression of R77Q compared to WT (*p* = 0.0326) (Figure 2C). We thus conclude that the R77Q variant is expressed at low levels relative to the WT isoform, although the reasons for this finding are still unclear. Despite the low-level expression of R77Q *vpr*, several key phenotypic differences will be presented herein that distinguish the Null mutant (no detectable *vpr* expression) from the R77Q mutant (weak, but detectable *vpr* expression), suggesting that expression level alone cannot explain the differences in phenotypes between these viruses.

### 3.4. Infection of HUT78 Cells with the R77Q Mutant Leads to an Increase in Apoptotic Cells

To test how polymorphisms in *vpr* affect the induction of apoptosis, we stained infected cells (MOI 0.01) with labeled Annexin V (for apoptosis) and fixable viability dye (FVD; for cells with a permeabilized membrane) and analyzed the results via flow cytometry at 3, 5, and 7 dpi (Figure 3). Annexin V binds to phosphatidylserine, which flips from the inner layer of the plasma membrane to the outer layer during the early stages of apoptosis [56]. Cells were also stained with FVD, which enters cells through porous plasma membranes and indiscriminately binds proteins. Since cells with a permeabilized membrane will allow Annexin V staining due to the penetration of the stain to the phosphatidylserine inside of the plasma membrane via porous membranes, we considered cells positive for both FVD and Annexin V as necrotic (“dead”), while cells positive for Annexin V alone were considered to be apoptotic (“dying”).

We observed a clear increase in apoptotic induction by the R77Q mutant in HUT78 cells (Figure 3). At every time point, the R77Q-infected population had significantly higher levels of apoptotic cells compared to all other samples (see Appendix A). At 7 dpi, the R77Q population had a mean level of 47.4 +/− 0.82% Annexin V single-positive cells compared to 2.79 +/− 0.14% in WT (*p* = 0.002), 6.94 +/− 0.42% in Null (*p* = 0.02), and 3.15 +/− 0.39% (*p* = 0.0003) in the uninfected population (Figure 3A,B). The Null virus-infected population also had a significant increase in the number of apoptotic cells at 7 dpi compared to the WT population (*p* = 0.02), but the effect was much lower as compared to R77Q. The number of apoptotic cells in the WT population was not significantly different from the uninfected population at any time point, suggesting that the *vpr* isoform in strain NL4-3 cannot effectively induce apoptosis in this model.

To explore whether production viral infection was required for the apoptotic phenotype, we repeated the above experiment but with the addition of p24 staining and gating. p24 is a major capsid protein of HIV-1 and is commonly used to detect productively infected cells. We found that after infection of HUT78 cells with the R77Q mutant, the p24+ population had high levels of Annexin V staining, with 49.3 +/− 0.9% detected (Figure 4). Infections with the WT and Null viruses showed low levels of p24+Annexin V+ cells, at 2.9 +/− 0.1% and 8.6 +/− 0.4%, respectively. The p24+Annexin V+ population was significantly higher following R77Q infection than with the other viruses (R77Q vs. WT, *p* = 0.0002; R77Q vs. Null, *p* = 0.0003). Similarly, the p24− population was significantly higher for apoptosis following infection with the R77Q (15.1 +/− 0.50%) than with WT (1.4 +/− 0.14%; *p* = 0.0004) or Null viruses (2.9 +/− 0.63%; *p* = 0.0005). These results suggest that productive infection enhances the rate of apoptosis after R77Q infection but that it is not required for the induction of apoptosis. Although the Null virus showed significantly fewer apoptotic cells than R77Q, the apoptotic induction was significantly higher than the WT virus in both the p24+ (*p* = 0.001) and p24− populations (*p* = 0.04) similar to the results shown in Figure 3B where all cells were analyzed. Similar to Figure 3, the WT virus had significantly more necrotic/FVD+ cells than either the R77Q (*p* = 0.0016) or Null viruses (*p* = 0.0073) when gating on p24, but when gating on p24− cells, no significant differences were detected between WT and other viruses. We also noted that the p24+ populations consistently showed significantly higher levels of both apoptosis and necrosis/FVD+ staining when compared to the p24− populations, regardless of viral strain. See Appendix A for all experimental results plus statistical analysis.

To determine whether this increase in apoptosis was detectable in other human T cell lines besides HUT78, we replicated the experiment in CEM cells. A similar apoptotic phenotype was observed at 3 dpi with a MOI of 0.01: 16.15 +/− 1.09% of the R77Q-infected population was apoptotic, which was significantly higher than those of the WT virus (4.17 +/− 1.07%; *p* = 3.7 × 10^−4^), Null virus (2.10 +/− 0.30%; *p* = 3.2 × 10^−5^), and uninfected cells (2.25 +/− 0.23%; *p* = 1.5 × 10^−5^) (see Appendix A, and Appendix A). Although the increase in apoptotic cells was seen in both cell lines, CEM cells showed the apoptotic phenotype at earlier time points compared to HUT78 cells.

### 3.5. Infection of HUT78 Cells with WT Vpr Leads to an Increase in Necrotic Cells

We also observed clear differences in the number of necrotic cells in each population, as defined by porous plasma membranes and staining with FVD. At 7 dpi, we observed significantly higher percentages of necrotic cells in all three infected populations compared to that of the uninfected control (Figure 3C, Data Set 2A). The percentage of dead cells at 7 dpi in the WT-infected population (25.4 +/− 1.4%) was significantly higher than in the R77Q (15.5 +/− 0.12%; *p* = 0.008) and Null populations (13.4 +/− 1.1%; *p* = 0.01) at 7 dpi (also significant for both at 5 dpi, see Appendix A). The lack of an FVD (−), Annexin V (+) population following infection with the WT virus suggests that this strain preferentially activates a necrotic pathway to kill cells. There was a significant difference in this dead cell population between R77Q and Null (*p* = 0.0018) but only at 5 dpi.

### 3.6. Primary Human CD4+ T Cells Show the Same Apoptotic Induction Following R77Q Infection

Since cell lines are immortalized, they may show different responses to apoptotic or necrotic signaling pathways compared to primary CD4+ cells following HIV-1 infection. We thus tested primary CD4+ T cells isolated from human umbilical cord blood for the induction of apoptosis, following infection with the WT and mutant viruses. Blood samples from four different donors were infected independently for 7 days to ensure reproducibility, and two donors were categorized as Caucasian donors while the other two were listed as non-Caucasian to ensure a level of diversity in subjects tested. As with the HUT78 and CEM cells, infection with the R77Q virus yielded significantly higher levels of apoptotic cells compared to all other samples (across all four patients). Patient 2 was most representative, and those results are shown in Figure 5. All patient data are shown in Appendix A: R77Q infection showed 21.60% +/− 0.85% apoptotic cells (Figure 5, with uninfected cells at 0.91% +/− 0.16% (*p* = 0.002 vs. R77Q), WT at 5.85% +/− 0.33% (*p* = 0.006 vs. R77Q), and Null at 2.84%+/− 0.46% (*p* = 0.001 vs. R77Q). We also observed that WT infection led to significantly more necrotic cells than any other experimental group (Figure 5C; See Appendix A for controls and representative dot blots with ancestry plots. Patient 3 is shown here; all patient data are shown in Supplementary Data Set S7): WT infection showed 36.62% +/− 1.65% necrotic cells, with uninfected cells at 4.90% +/− 0.99% (*p* = 0.004 vs. WT), R77Q at 14.92% +/− 1.67% (*p* = 0.0001 vs. WT), and Null at 14.41% +/− 1.38% (*p* = 0.017 vs. WT). We thus conclude that apoptotic induction by the R77Q mutant can be detected following infection with replication-competent HIV-1 in both primary T cells and in multiple T cell lines and that WT infection also leads to more necrotic cell death in primary cells.

### 3.7. TUNEL Staining Confirms That Only the R77Q Mutant Induces Apoptosis in HUT78 Cells

To confirm that the AnnV+ populations from the previous experiment were truly apoptotic, infected HUT78 cells were stained for DNA fragmentation—a classic sign of late-stage apoptosis [56]. The TUNEL (terminal deoxynucleotidyl transferase-dUTP nick end labeling) assay, analyzed by flow cytometry, showed minimal signs of DNA fragmentation in the uninfected, WT, and Null mutant-infected populations at 5 dpi. The R77Q-infected population was 17.2 +/− 2.0% positive for TUNEL, while all the other populations had minimal TUNEL+ cells. Statistical analysis showed that this increased percentage in TUNEL+ cells in the R77Q-infected population was significant compared to that of the uninfected (*p* = 0.007), WT virus (*p* = 0.007), or Null-infected cells (*p* = 0.007) (Figure 5B and Figure 6A, Appendix A). No other samples were significantly different than one another, although WT vs. uninfected approached significance (*p* = 0.057). The percentage of R77Q-infected cells positive for Annexin V was higher than the percentage of cells positive for TUNEL; this is likely due to the fact that phosphatidylserine membrane flipping occurs earlier during the apoptotic process than DNA fragmentation [56], allowing Annexin V to label both early- and late-stage apoptotic cells. Together, these data show that the R77Q mutant primarily kills cells through the induction of apoptosis, while the WT virus primarily causes death through different, non-apoptotic pathways.

### 3.8. G_2_ Cell Cycle Arrest Is Significantly Increased in Cells Infected with the R77Q Mutant

To determine whether these mutants differed in their capacity to trigger G_2_ cell cycle arrest, we used flow cytometry to measure the cell cycle progression of infected cells. HUT78 cells were infected at 0.1 MOI with the WT, R77Q, or Null viruses, then stained at 7 dpi with propidium iodide and with antibodies to the p24 antigen, and analyzed through flow cytometry. The p24+ populations were gated during analysis to focus specifically on cell cycle arrest in productively infected cells. Figure 7 shows a significant mean increase in the detection of cells in the G_2_ phase in all infected samples compared to the uninfected control (19.2 +/− 1.35% in G_2_): (WT = 26.0 +/− 0.75%, *p* = 0.01); R77Q = 40.4 +/− 3.18%, *p* = 0.01; Null = 26.0 +/− 2.19%, *p* = 0.03; see Supplementary Data Set S4). R77Q showed the highest percentage of cells arrested in G_2_, which was significantly higher than those of both the WT and Null viruses (vs WT *p* = 0.02; vs. Null *p* = 0.01). This finding was expected since it has been previously reported [9,57]. Since the Null virus also showed higher levels of cells in the G_2_ phase as compared to uninfected cells, some other factor besides *vpr* may promote G_2_ arrests during HIV infections. We also noted an increased accumulation of cells in the S phase following infection with either WT, R77Q, or Null viruses as compared to uninfected cells, as reported previously [58]. However, R77Q did not show a statistically significant difference in S phase accumulation compared to the uninfected population (*p* = 0.07), possibly because an increase in one population percentage must result in corresponding decreases in other populations, and cells in the G_2_ phase were much higher in these samples. A summary of the raw data, including an analysis of S and G1 populations, is found in Appendix A. These results support studies that show that the R77Q mutant enhances G_2_ cell cycle arrest [9]. Early experiments used the same 0.01 MOI as reported for other figures, but no significant differences were detected in the cell cycle. The p24 stains showed that relatively few cells were infected at the time points analyzed, so we moved to use MOI 0.1, and we gated on p24+ cells and began to find statistically significant results. An analysis was also performed of p24− cells, but there was no significant difference between WT *vpr* and the mutants or any infected cell samples compared to uninfected controls (data not shown). This result suggests that the productive infection of cells with HIV-1 is required for modifications of the cell cycle.

### 3.9. HUT78 Infection with the R77Q Mutant Results in Lower Levels of Pro-Inflammatory Cytokines as Compared to WT Virus

We next sought to determine whether differences could be detected in inflammatory markers in the supernatants of HUT78 cells infected with our various strains of HIV-1 bearing different *vpr* genes. Tumor necrosis factor (TNF) and interleukin 6 (IL-6) represent pro-inflammatory cytokines associated with AIDS progression and were quantified using a cytometric bead array. The mock infection resulted in low levels of TNF (15.2 +/− 2.0 pg/mL) at 5 dpi. The WT-infected cells showed significantly higher secretions of TNF (197.3 +/− 19.1 pg/mL) than those of the mock (*p* = 4 × 10^−6^), R77Q (57.3 +/− 7.7 pg/mL; *p* = 8 × 10^−5^) and Null (89.3 +/− 18.2 pg/mL; *p* = 6.7 × 10^−4^) infected populations (Figure 8A). The R77Q and Null infections had more TNF than the mock infection (*p* = 1.5 × 10^−4^ and *p* = 4.7 × 10^−3^), but there was no significant difference between the TNF expression from R77Q- vs. Null-infected cells (*p* = 0.07). These results suggest that HIV-1 infection generally upregulates TNF expression in HUT78 cells, but that the R77Q and Null mutants fail to show this phenotype, and these two mutants behave similarly in this regard.

When IL-6 expression was similarly measured, mock infections resulted in low amounts of IL-6 expression (9.7 +/− 1.1 pg/mL) when compared to those of WT (22.0 +/− 3.4 pg/mL; *p* = 0.004), R77Q (13.9 +/− 3.3 pg/mL; *p* = 0.02), and Null infections (18.6 +/− 2.7 pg/mL; *p* = 0.009) (Figure 8B), again suggesting that HIV-1 infection generally activates the expression of IL-6. The difference between WT and R77Q was also significant (*p* = 0.02), although the expression was not statistically different between R77Q and Null (*p* = 0.14) or between WT and Null (*p* = 0.23). Taken together, these results suggest that infection with WT *vpr* induces a significantly stronger pro-inflammatory response than infection with the R77Q allele. This finding was partially replicated when comparing the WT infection to the Null virus where the difference was significant for the TNF expression but not for the IL-6 expression.

## 4. Discussion

In this study, we used mutant strains of (replication-competent strain NL4-3) HIV-1 with point mutations in the *vpr* gene to gain a greater understanding into why the R77Q mutant is potentially attenuated for AIDS progression. We noted several phenotypes that differ between the R77Q *vpr* mutant and WT *vpr*, leading to four main conclusions: (1) the NL4-3-based R77Q *vpr* mutation enhances the pro-apoptotic activity of HIV-1 in multiple human T cell lines and in primary human T cells; (2) the NL4-3-based WT and Null viruses kill both CD4+ T cell lines and primary cells via a necrotic, non-apoptotic pathway; (3) the R77Q *vpr* mutation significantly enhances G_2_ cell cycle arrest in HUT78 cells; (4) the R77Q mutant fails to induce the expression of pro-inflammatory cytokines in HUT78 cells to the extent seen by the WT virus. Our studies were conducted with replication-competent HIV-1 in human CD4+ T cells, and we believe that discrepancies that have been reported in the past may be in part due to (1) the use of *vpr* expression plasmids vs. replication-competent virus (which expresses other pro-apoptotic proteins such as Env and Nef) and (2) the use of unnatural HIV-1 target cells to study these phenotypes. We also believe that the R77Q mutation alone is not sufficient to confer these phenotypes in cultured cells, but that other sequences in the gene are also critical (see below).

We found significant differences between R77Q and WT/Null viruses in terms of their death pathways, with a strong induction of apoptosis that was unique to R77Q infection. There is conflicting evidence regarding the mechanism by which HIV kills infected cells. Many studies have indicated that HIV-1 infection triggers apoptosis [59,60], while others have provided evidence that it induces a non-apoptotic form of cell death [19,20]. In some specific circumstances, it has been shown to induce pyroptosis [61]. Our results show that WT NL4-3 induces necrosis during infection, while a point mutation resulting in R77Q causes a shift to a pro-apoptotic phenotype [62,63,64]. This result was consistent across both cell lines and primary cells and via both Annexin V and TUNEL staining approaches, which detect apoptosis by very different mechanisms. Although productive infection (gating on p24+ cells) strongly correlated with the apoptotic phenotype, the lack of detectable productive infection (p24− cells) also showed an increase in apoptosis. Our approach cannot distinguish between a lack of infection, or abortive infection, in the p24− cell population. This is relevant to previous work by others that has shown a link between abortive reverse transcription and apoptotic induction [35]. We also recognize that other experiments included here could be repeated in primary CD4+ T cells besides just the induction of apoptosis, which may provide additional insight into viral replication kinetics, cell cycle arrest, and pro-inflammatory cytokine production in a more relevant cell model.

We acknowledge that a number of previous studies have failed to detect increased apoptosis in cells following exposure to R77Q *vpr* [22,39] and/or that WT *vpr* can induce apoptosis more than R77Q can do [65]. The difference between our observed R77Q apoptotic phenotype and those previously reported is also likely a result of different experimental systems. Previous studies that associated R77Q *vpr* with a normal or decreased capacity for apoptosis were performed in HeLa cells transduced with R77Q *vpr*-expressing plasmids [22] that may express supra-physiological levels of *vpr* or in Jurkat cells infected with the VSV-G pseudotyped virus [35,61]. Pseudotyped HIV lacks Env, which is well documented to induce apoptosis [39,65], and it is possible that R77Q Vpr requires Env activity to induce apoptosis. Our experimental system more closely resembles the natural conditions of infection since we have studied *vpr* function in CD4+ T cells and in the context of replication-competent HIV, which includes the apoptotic functions previously ascribed to *env*, *vif*, *nef*, etc. [66,67,68]. As a cancer cell line, HUT78 cells are relatively resistant to apoptosis; this resistance is a result of null mutations in the *p53* gene [69,70]. However, since *vpr*-induced apoptosis is p53-independent [24,71], we believe that R77Q *vpr* is a *bona fide* pro-apoptotic mutation.

Our analysis of the cell cycle showed that R77Q infection causes a significant arrest in the G_2_ phase, a finding previously reported [72], but again there are discrepancies in the literature in regards to the ability of the R77Q mutant to impact this phenomenon [22]. Several previous studies have concluded that there is a link between the arrest of the cell cycle and induction of apoptosis, including in T cell lines and in primary T cells [26,71], similar to our findings where R77Q induces these changes significantly more than the WT virus. It was suggested by Andersen et al. that apoptosis results from prolonged G_2_ arrest [22], and Zhu et al. showed that caffeine exposure inhibits both the G_2_ arrest and apoptotic phenotypes normally induced by HIV-1 *vpr* [73]. Caffeine inhibits both ATM and ATR, which regulate the cell cycle, providing evidence that cell cycle arrest and apoptosis are indeed linked.

We detected a 6-fold reduction in the detection of *vpr* protein in R77Q-infected samples as compared to the WT virus by immunoblotting (Figure 2). In our analysis of the replicative capacity of each virus, by a Q-RT-PCR of viral load, we showed that both the R77Q and the Null mutant behaved very similarly to the WT virus in replication rate, suggesting that these mutations do not impact viral replication efficiency in an in vitro setting in T cell lines. This decreased level of *vpr* expression from the R77Q virus was consistently detected with different MOI (data not shown),and should not be due to a change in antibody binding to the target protein because the polyclonal antibody used for detection was generated against the first 50 amino acids of *vpr*, and the R77Q mutation falls outside of this region. Thus, we wondered whether the apoptotic and cell cycle arrest phenotypes could be explained by a simple lack of *vpr* expression. However, our use of the Vpr Null mutant, which never showed a detectable *vpr* expression by immunoblotting (Figure 2), showed a weak induction of apoptosis and no detectable G_2_ cell cycle arrest phenotypes (note that the Null virus was similar to R77Q in terms of reduced pro-inflammatory cytokine production). Thus, we conclude that decreased *vpr* expression alone cannot explain the correlation between low or non-existent *vpr* expression and multiple observed phenotypes. However, the weak induction of apoptosis from the non-*vpr*-expressing Null virus and the strong induction of apoptosis by a weakly *vpr*-expressing R77Q mutant suggests that there may be a correlation between poor or undetectable *vpr* expression and the apoptotic and cytokine production phenotypes.

It is possible that low levels of *vpr* expression give a different phenotype as compared to the high expression or a complete lack of expression; our experiments did not address this possibility. Other mutations in *vpr* have previously shown poor expressions by immunoblot, such as an isolate with 77A [74] or another with 64P. The 64P strain also failed to induce apoptosis or G_2_ arrest, although it should be noted that it was also 77R at that position [72]. This same study performed a longitudinal analysis of *vpr* sequences correlated with apoptotic and G_2_ arrest abilities and suggests that other mutations, beyond just 77Q, contribute to or detract from these phenotypes. This may explain, at least in part, the discrepancies previously reported for the R77Q mutation and its ability to trigger apoptosis and G_2_ arrest. We do not believe that 77Q alone is sufficient to always confer the apoptotic or G_2_ arrest phenotypes, as evidenced by a study by Jacquot et al. that showed a failure by some 77Q-containing viruses to induce G_2_ arrest (e.g., strain LTNP-04-2 Q77), or showed a reduced apoptotic phenotype relative to other 77Q-containing viruses. Other amino acids in strain LTNP-96 in that study appear to be important, although 77Q alone seems to behave very differently compared to 77R in that genetic background [72]. Note that Lum et al., whose results differ from ours in terms of whether WT or R77Q viruses are more apoptotic, used strain HxBRUR in their study [39], which expresses *vpr* with N28; our strain NL4-3 has *vpr* S28. S28 is one mutation that seems to be important to both the apoptotic and G2 arrest phenotypes because virus strains LTNP-86 and LTNP-04-1 (both are N28) were mutated to have R77Q, but they failed to show an increase in either these phenotypes [72]. Other studies that fail to show R77Q-induced apoptosis may also have N28.

AIDS progression is strongly correlated with inflammatory markers and chronic immune activation, and upregulated expression of pro-inflammatory cytokines such as TNF and IL-6 has been documented both in vitro and in vivo during HIV-1 infection [75,76,77]. While the expression of IL-6 can be reduced after HAART therapy, TNF expression can persist at elevated levels even when viral replication is inhibited [78]. Previous research has found that the de novo production of *vpr* in HIV-1 infected cells triggers the release of TNF, which is a pro-inflammatory cytokine. Interestingly, the authors noted a correlation between the inability to cause G_2_ arrest and lower TNF expression by certain *vpr* alleles [79]. The cytometric bead array results showed that the R77Q mutation resulted in a lower expression of TNF when compared to that of WT, but there was no statistical difference between Null and R77Q (89.3 vs. 57.3 pg/mL respectively; *p* = 0.07) infection. Similarly, the R77Q infection failed to induce high levels of IL-6 as seen with the WT infection (*p* = 0.02), and there was no statistical difference between R77Q and Null (*p* = 0.14). These results suggest that a low expression of *vpr* cannot fully explain the differences seen in pro-inflammatory cytokine production. These results show that the environment within a WT HIV-1 infected cell results in an increase in production/secretion of multiple pro-inflammatory cytokines vs. infection with the LTNP-associated R77Q virus. This, along with the results showing that R77Q pushes towards apoptosis more than WT, suggests that R77Q causes less inflammation than WT in vitro. If also true in vivo, this decrease in inflammation could contribute to the LTNP phenotype associated with the R77Q mutation.

## 5. Conclusions

Although the R77Q mutation was linked to LTNP in 2004, there is still a poor understanding of any associated mechanisms to AIDS progression. A tendency to induce pro-inflammatory death could potentially lead to a faster and/or more potent chronic activation of the immune system, which has been hypothesized to lead to AIDS progression [80]. Several of our results indicate that an R77 virus is more prone to induce necrosis and inflammation than a Q77 virus in our model system. It is also known that apoptosis does not enhance a systemic inflammation [81] and therefore could contribute to a delay of the chronic activation of the immune system that leads to AIDS. If the R77Q mutation influences host cells to die primarily by apoptosis, this could potentially explain the associated LTNP phenotype. Although R77Q is highly cytotoxic and replicates at a similar rate to the WT virus, the death pathway followed by infected cells could potentially alter the host immune response during a chronic HIV-1 infection. Future research is needed to determine the mechanisms that can explain how polymorphisms in *vpr* can impact pro- and anti-inflammatory death pathways.

## Figures and Tables

**Figure 1 viruses-16-01642-f001:**
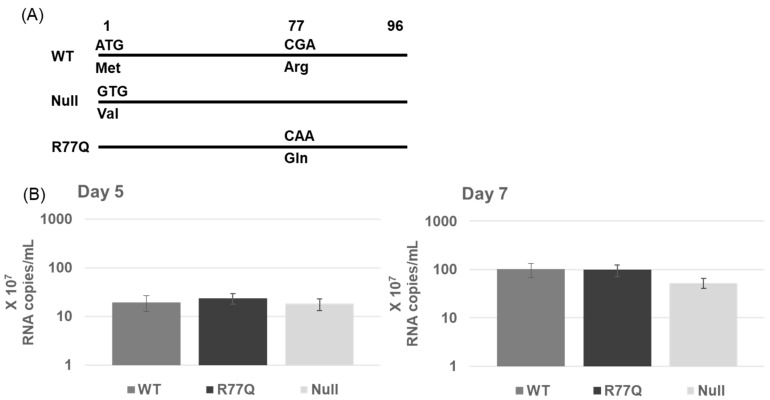
(**A**) Nucleotide and amino acid differences in the *vpr* gene for various strains used in this study. (**B**) *vpr* mutations have minimal impact on the replication capacity of HIV-1 in HUT78 cells. WT, R77Q, or Null mutants were used to infect HUT78 cells at MOI 0.01. At 3, 5, and 7 dpi, supernatants were analyzed for viral load through Q-RT-PCR. See Appendix A for complete statistical analysis.

**Figure 2 viruses-16-01642-f002:**
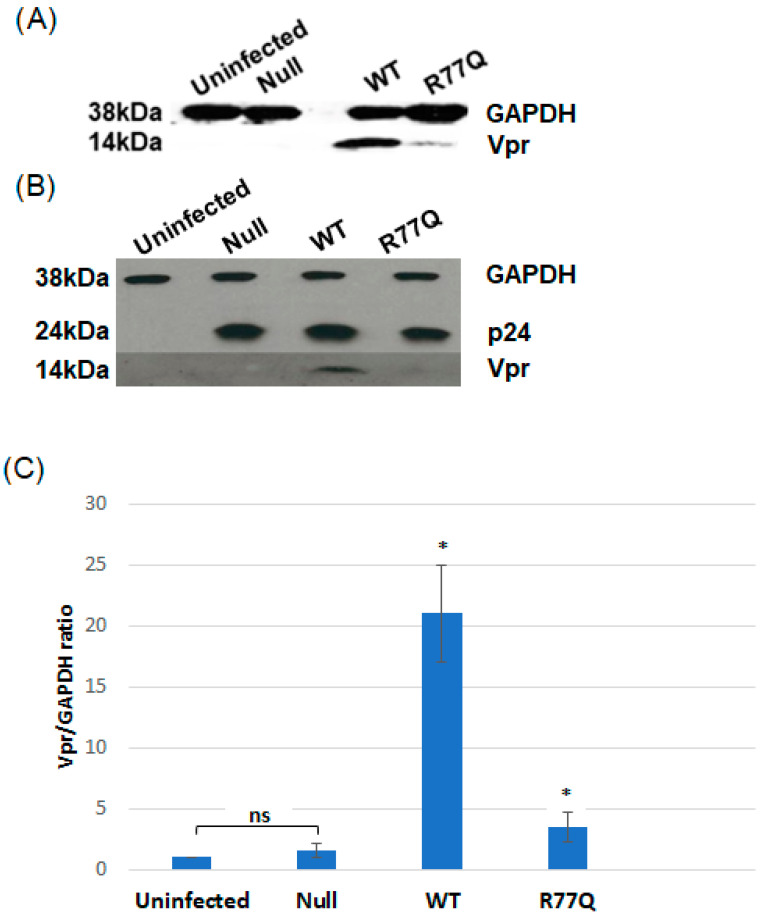
Differences in nucleotide and amino acid sequences between *vpr* mutants and confirmation of gene expression by immunoblot. (**A**) Immunoblotting confirmed *vpr* expression in HUT78 cells infected with WT virus or the R77Q mutant, while no *vpr* expression was detected in the Null mutant or in uninfected cells. GAPDH was used as a loading control. (**B**) p24 was also used as a loading control for differing viral replication kinetics and/or protein production. (**C**) Bar graph indicating the quantitative analysis of the relative western blot bands expressed as a ratio to the uninfected control. Data are representative of three independent experiments, and error bars indicate SE. Asterisk * indicates that there was a statistically significant difference when compared to all other conditions (*p* ≤ 0.05) whereas n.s. signifies no statistical significance (*p* > 0.05).

**Figure 3 viruses-16-01642-f003:**
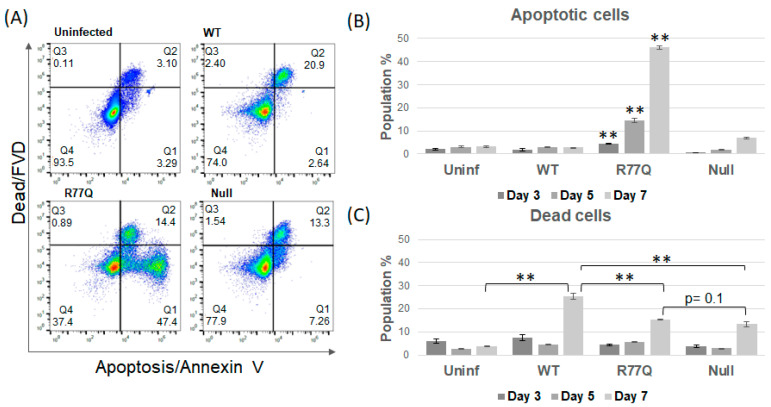
The R77Q *vpr* mutant triggers different death pathways in HUT78 cells than WT and Null. WT, R77Q, and Null HIV *vpr* mutants were used to infect HUT78 cells at MOI 0.01, and cell samples were analyzed at 3, 5, and 7 dpi. Apoptosis was detected by Annexin V staining and fixable viability dye (FVD) was used to detect dead cells via penetration of the dye across a porous plasma membrane. Thus, all FVD+ cells are expected to also bind Annexin V but not due to phosphatidylserine flipping to the outer membrane. (**A**) Representative dot plots from samples collected at 7 dpi. (**B**) Apoptotic cells (positive for Annexin V only). (**C**) Dead cells (all cells positive for FVD). At 3 dpi, no statistical differences were detected between any samples. At 5 dpi and 7 dpi, all samples were statistically different from each other, with the sole exceptions being Null and uninfected at 5 dpi and R77Q and Null at 7 dpi. Data are representative of three independent experiments, and error bars indicate SE. ** *p*-value ≤ 0.01; statistics shown focus on 7 dpi. See Appendix A for complete datasets and statistical analysis. For the flow cytometry dot plots, the x-axis runs from 10^0^ to 10^7^ in increments of 10, while the y-axis runs from 10^0^ to 10^8^ in increments of 10.

**Figure 4 viruses-16-01642-f004:**
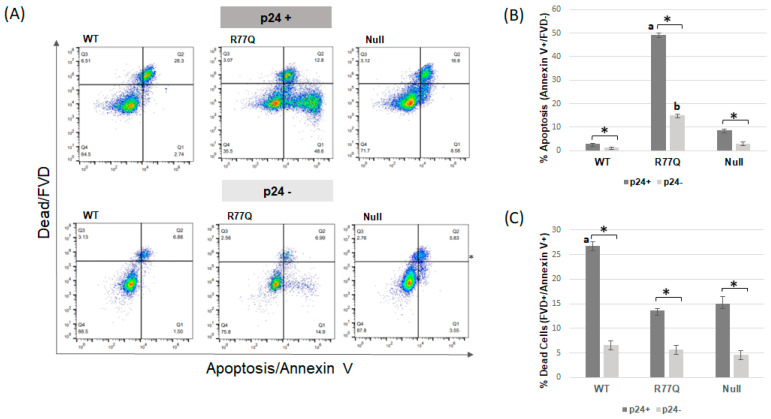
HUT78 cells productively infected with R77Q virus are highly apoptotic, but non-productively infected cells also showed increased apoptosis. WT, R77Q, and Null HIV *vpr* mutants were used to infect HUT78 cells at MOI 0.01, and cell samples were analyzed at 7 dpi. Apoptosis was detected by Annexin V staining, fixable viability dye (FVD) was used to detect dead cells, and an antibody to the HIV-1 p24 protein was used to detect productively-infected cells. (**A**) Representative dot plots from samples collected on 7 dpi, with gating first performed on the p24+ or p24− populations. (**B**) Apoptotic cells (positive for Annexin V only, after first gating on p24+ or p24− populations). (**C**) Dead cells (all cells positive for FVD, after first gating on p24+ or p24− populations). Error bars indicate SE. * *p*-value ≤ 0.05. See Appendix A for complete datasets and statistical analysis. ^a^ indicates that the labeled population was significantly different than all other virus strains when comparing within the p24+ populations; similarly, ^b^ indicates that the labeled population was significantly different than all other virus strains within the p24− populations. For the flow cytometry dot plots, the x-axis runs from 10^0^ to 10^7^ in increments of 10, while the y-axis runs from 10^0^ to 10^8^ in increments of 10.

**Figure 5 viruses-16-01642-f005:**
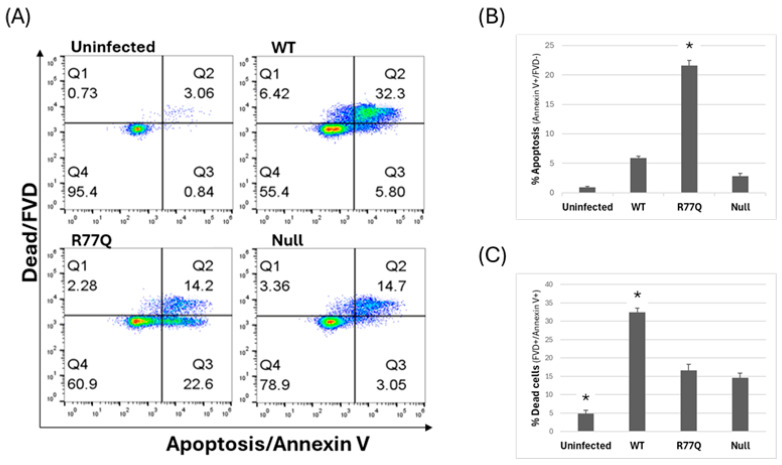
Primary CD4+ T cells become apoptotic after infection with the R77Q mutant. CBMC-derived CD4+ T cells were infected with WT NL4-3, R77Q, or Null HIV *vpr* mutants in triplicate at MOI 0.01 and analyzed for apoptosis at 7 dpi using the Annexin V/Fixable viability dye staining method. Shown are results from a representative sample from a non-Caucasian CBMC donor (Patient 2). Similar results were obtained from three additional donors (two Caucasian and one non-Caucasian). See Appendix A for results from additional patients and for statistical analysis. (**A**) Representative dot plots of Annexin V/FVD for uninfected and WT NL4-3, R77Q, and Null HIV *vpr*-infected CBMC-derived CD4+ T cells. (**B**) Bar graph representing the proportion of apoptotic cells (Annexin V+ only) for patient 2 alone. CD4+ T cells were gated from CBMCs, and singlet discrimination was performed before gating on Annexin V/FVD guided by single color controls and an unstained control (Appendix A). R77Q showed a statistically significant difference when compared to all other groups. All groups were statistically different from each other, except for the comparison between Null and uninfected (*p* = 0.085) and Null and WT (*p* = 0.063). (**C**) Bar graph representing the proportion of dead cells (FVD+ and Annexin V+). The gating strategy was performed as described in B above. WT and uninfected were statistically different when compared to all other groups. All other groups were statistically different from each other except for R77Q compared to Null (*p* = 0.466). Error bars indicate standard error while * indicates significant difference compared to all other groups (*p* < 0.05). For the flow cytometry dot plots, the x-axis runs from 10^0^ to 10^6^ in increments of 10, while the y-axis runs from 10^0^ to 10^6^ in increments of 10.

**Figure 6 viruses-16-01642-f006:**
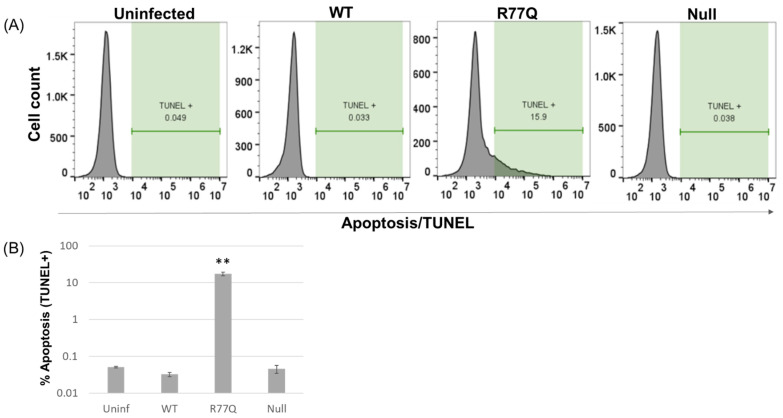
R77Q-enhanced apoptosis confirmed by TUNEL stain. WT, R77Q, and Null HIV *vpr* mutants were used to infect HUT78 cells at MOI 0.01, and cell samples were stained by the TUNEL assay and analyzed at 5 dpi. (**A**) Representative dot plots from 5 dpi. (**B**) Bar graph representing the TUNEL-positive cells of each sample. Asterisks show statistical differences between R77Q and all the other samples. Data are representative of three independent experiments, and error bars indicate SE. ** *p*-value ≤ 0.01. See Appendix A for complete statistical analysis.

**Figure 7 viruses-16-01642-f007:**
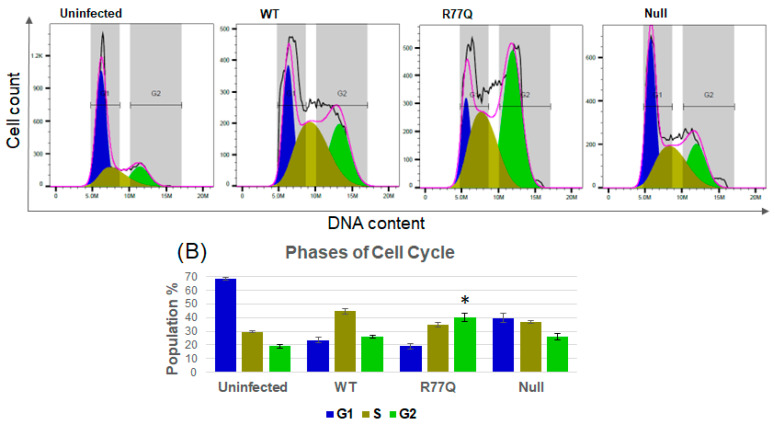
Infection of HUT78 cells with the R77Q *vpr* mutant enhances G_2_ cell cycle arrest. WT, R77Q, and Null HIV *vpr* mutants were used to infect HUT78 cells at MOI 0.1, then stained with propidium iodide to detect relative DNA content, and analyzed by flow cytometry. Cells were first gated on p24+ populations prior to cell cycle analysis in order to focus on productively infected populations. (**A**) Representative histograms from 7 dpi. (**B**) Percentage of p24+ cells in G_2_ phase. The asterisk shows a statistical difference between R77Q and all other samples. Data are representative of three independent experiments, and error bars indicate SE. * *p*-value ≤ 0.05. See Appendix A for complete statistical analysis. Colors in the bar graphs match the various populations in the histograms above. The x-axis ends at 21M for each sample. The y-axis ends at 1.4K for Uninfected, at 500 for WT, at 560 for R77Q, and at 640 for Null.

**Figure 8 viruses-16-01642-f008:**
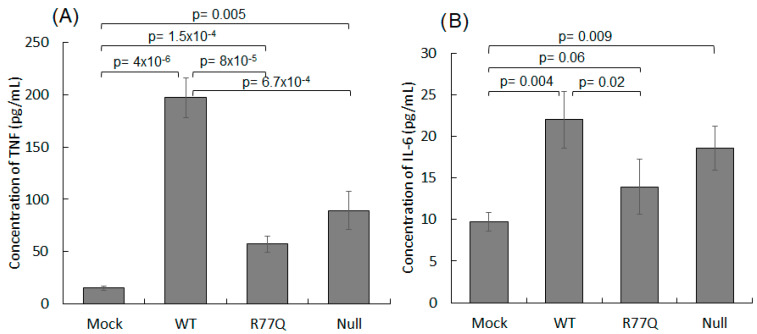
Expression of TNF and IL-6 are strongly upregulated after infection with WT virus, but not with the R77Q mutant. WT, R77Q, and Null viruses were used to infect HUT78 cells at MOI 0.01, and then at 5 dpi, supernatants were collected and assayed for TNF (**A**) and IL-6 (**B**) production by cytometric bead array. Data are representative of three independent experiments, and error bars indicate SE. See Appendix A for complete statistical analysis.

## Data Availability

All data generated or analyzed during this study are included in this published article.

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
