# Peer review of "The HIV-1 vpr R77Q Mutant Induces Apoptosis, G2 Cell Cycle Arrest, and Lower Production of Pro-Inflammatory Cytokines in Human CD4+ T Cells"

_viruses, 2024, doi:10.3390/v16101642_

Round 1
Reviewer 1 Report
Comments and Suggestions for Authors
The auhtos generate competent HIV-1 virus strains carrying VPR with R77Q mutation to test the effects of mutant VPR on cellular functions. Comparing with wild type (WT) HIV-1 virus (NL4-3) which caused necrotic cell death and stimulated inflammatory cytokines, including IL-6 and TNFa. The virus strain with mutant VPR induced apoptosis and G2 phase arrest and lower levels of inflammatory cytokines. Moreover, the virus stain without vpr gene also induced necrotic cells death like wild type strain. he authors concluded that HIV-1 with R77Q VPR tends to cause apoptosis and less inflammation, probably contributing to Long-Term Non-Progressors (LTNP) in some patients with HIV infection. However, authors need to address several concerns before consideration of publication.
1. The VPR protein of HIV virus have been shown to induce apoptosis when expressing at high levels (Mbita, virus 2014; Varbanov, AIDS Reviews 2006; Muthumani, JBC 2002.); however, R77Q VPR can impair apoptosis in T lymphocytes (Lum, JCI 2003; Mologni, AIDS 2006; Jacquot, PLoS One 2009). The conclusion in this study is contradict to these findings.
2. In Fig. 4 and 5, cells death should be analyzed in cells with HIV infection, not in whole cell population. For example, p24 gating should be applied to avoid other artificial effects.
3. In Fig. 6A, both apoptotic cells (TUNEL+ cells) and FVD+ dead cells increased in cells with R77Q virus infection. But WT and VPR null virus did not induce FVD+ cell death. These data are different from observations in Fig 4 and 5, in which WT and VPR null virus caused FVD+ cell death, the major phenotypes authors claimed.
4. In Figure 2B, both low level of p24 and VPR were found in HUT78 cells infected with R77Q virus. This is probably due to different infection efficacy among various virus strains. If so, analyzing cells infected with virus, not whole cells population, should be well defined. Another possibility is that R77Q virus somehow have low efficiency for producing virus. Low levels of R77Q virus in cells comparing with WT and VPR null virus may account for the phenotypes observed.
5. Correcting typos. For examples, Line 204, “3. Results”.
Comments on the Quality of English LanguageSome typos should be corrected (see abve)
Reviewer 2 Report
Comments and Suggestions for Authors
This study by Berges and colleagues has explored the phenotypic effects of the Vpr R77Q mutant vis-à-vis the wildtype or a Vpr null mutant in T cells infected with replication-competent NL4-3 versions of these viruses. The T cells utilized in the study were primarily the HUT78 cell line with some confirmatory experiments conducted using CEM and cord blood-derived primary CD4+ T cells. The authors report that infection of T cells with the R77Q Vpr virus elicited a significant enhancement in apoptosis and G2 cell cycle arrest over the wildtype, despite the fact that these two phenotypes were previously well-established to be associated with Vpr. This is also despite a drastic reduction in Vpr expression observed with R77Q in HUT78 cells at 7 days post-infection. In addition, R77Q Vpr is reported to cause a statistically significant reduction in the production of TNF and IL-6 by infected cells. Based on these data, the authors have proposed that the preferential entry of R77Q Vpr-infected cells into the apoptotic pathway is associated with decreased inflammation that may in turn contribute to long-term non-progression to AIDS.
The manuscript is well-written and some of the experiments were well-conducted including the flow cytometric assays carried out to differentiate late-stage apoptotic and necrotic cells. However, other aspects of the study have a number of major concerns that need to be addressed or clarified. Firstly, infection data comparing the replication capacity of R77Q, wildtype and null Vpr viruses following infection of the cord-derived primary CD4 T cells are missing. Without knowing the extent to which these primary cells are infected with these viruses, it becomes difficult to interpret the data shown in Figure 5. Also, cell cycle arrest and cytokine measurements in infected primary CD4 T cells similar to those done with the cell line are missing. Secondly, there is inconsistency in making the determination on whether to focus on the infected fraction of cells in assessing effects of R77Q. In one instance in Figure 7 infected cells were first sorted based on p24 staining prior to cell cycle analysis, whereas assessment of apoptosis versus necrosis in Figures 4 and 5 was based on the entire cell population. Why did the authors choose not to perform p24 staining in tandem with the annexin V/FVD co-staining in order to identify the infected cells? This raises the question of the extent to which the cell death observed is occuring solely in the infected subset versus a bystander effect. Thirdly, while it was previously shown that Vpr is a pro-apoptotic viral factor with its apoptosis-inducing mechanism clearly delineated, this study presents evidence in Figures 4 and 5 contrary to these findings by suggesting minimal apoptosis caused by the wildtype Vpr virus. Thus, by demonstrating minimal apoptosis due to wildtype Vpr, the authors fail to provide clarity on the quantitative extent to which the R77Q virus is more superior at inducing apoptosis. There is also the confusing statement made in Lines 525-529 and elsewhere in the text that R77Q Vpr is not sufficient to confer the apoptotic phenotype; this is inconsistent with the authors’ interpretation of the results in Figures 4 and 5.
Specific comments:
1. This reviewer fails to understand the presentation of Figure 1 as a stand-alone figure especially considering that the mutant Vpr NL4-3 plasmids were not generated in this study but rather were a gift from someone else. This scheme could be included as a figure set with the data presented in Figure 2.
2. The immunoblotting results shown in Figure 3 which were done with HUT78 cells at Day 7 pi clearly show a profound reduction in Vpr expression due to R77Q. However, at Day 3 pi of the HUT78 cells where the release of the R77Q Vpr virus was found to be about 10-fold higher than that of wildtype virus, how do the expression levels of the mutant protein compare to that of the wildtype? Why did the authors choose not to examine Vpr protein expression at Days 3 and 5 pi?
3. If the immunoblotting experiments with representative data presented in Figure 3 were done ‘across many experiments’ as stated in Lines 243-245, the authors should consider providing an accompanying graph and statistical data quantifying the reduction in Vpr expression due to the R77Q mutation. As indicated above, also clarify whether this trend is also observed at Days 3 and 5 pi, preferably using p24 as the normalization control for the infection assuming that its expression is unaffected by the Vpr mutations.
4. Please comment on whether the reduced pro-inflammatory phenotype observed with the R77Q mutant at 5 dpi in Figure 8 is attributable to the reduced expression of Vpr? Why were these cytokines only measured 5 dpi and not also at 3 and 7 dpi?
5. In Figure 2, where infection is measured by quantifying virus release into culture media, a graph of Day 3 pi viral load data for the WT, R77Q and null Vpr viruses is missing. How does the infection with R77Q compare to that of the null virus at Day 3 pi?
6. The legend heading for Figure 2 is inaccurate since viral load of the R77Q virus at Day 3 pi was found to be 10-fold higher than that of the wildtype.
7. The authors indicate that the flow data presented in Supplementary Fig. 1 for the CEM cells are representative of 6 experiments. However, it is unclear why graphed data similar to the ones shown in Figures 4B and C for the HUT78 are not shown. Moreover, the representative data shown for the CEM cells are only from 3 dpi in contrast to those shown for HUT78. Why not present the overall data for CEM cells at Days 3, 5, and 7 pi?
8. Why did the authors choose to use a 10-fold higher MOI (0.1) to infect cells for cell cycle analysis rather than sticking with the 0.01 MOI used for all of the other experiments in the study?
9. Infection data comparing the replication capacity of R77Q, wildtype and null Vpr viruses following infection of the cord-derived primary CD4 T cells are missing. Without knowing the extent to which these primary cells are infected with these viruses, it becomes difficult to interpret the data shown in Figure 5. Also, cell cycle arrest and cytokine measurements in infected primary CD4 T cells similar to those done with the cell line are missing.
10. The authors should consider examining the protein expression of wildtype and R77Q Vpr in the infected primary T cells to confirm the observation made with the HUT78 cell line shown in Figure 3.
11. There is inconsistency in making the determination on whether to focus on the infected fraction of cells in assessing effects of R77Q. In one instance in Figure 7 infected cells were first sorted based on p24 staining prior to cell cycle analysis, whereas assessment of apoptosis versus necrosis was based on the entire cell population. Why did the authors choose not to perform p24 staining in tandem with the annexin V/FVD co-staining in order to positively identify the infected cells? This raises the question of the extent to which the cell death observed is occuring solely in the infected subset versus a bystander effect.
12. While it was previously shown that Vpr is a pro-apoptotic viral factor with its apoptosis-inducing mechanism clearly delineated, this study presents evidence in Figures 4 and 5 contrary to these findings by suggesting minimal apoptosis caused by the wildtype Vpr virus. Thus, by demonstrating minimal apoptosis due to wildtype Vpr, the authors fail to provide clarity on the quantitative extent to which the R77Q virus is more superior at inducing apoptosis. There is also the confusing statement made in Lines 525-529 and elsewhere in the text that R77Q Vpr is not sufficient to confer the apoptotic phenotype; this is inconsistent with the authors’ interpretation of the results in Figures 4 and 5.
Other comments:
1. Is it appropriate to refer to a T-cell line like HUT78 or CEM cells as ‘helper T cells’?
2. On the first page, the number, letter and symbol notations for institutional affiliations, author correspondence etc are misaligned.
Comments on the Quality of English Language
1. Please correct typos such as in Line 208 'CXRC4'.
Round 2
Reviewer 1 Report
Comments and Suggestions for Authors
The manuscript is significantly improved.
Author Response
There were no comments to be addressed from Reviewer 1
Reviewer 2 Report
Comments and Suggestions for Authors
1. In the renumbered Figure 4 (formerly Figure 5) the authors show that upon infection of cord blood-derived primary CD4+ T cells, the R77Q Vpr mutant virus induces a significantly higher level of apoptosis compared to the WT or null Vpr virus. My reviewer request was NOT for additional experiments, but only to show the extent of infection of these primary T cells with each of these viruses for experiments already conducted and whose data are shown in 5A-C. Similar to the data shown in Figure 1 for infection of the HUT78 cell line, what is the replicative capacity or extent of infection of the WT, R77Q and null virus in the primary T cells? It cannot be assumed that it will be the same as that observed in the cell line model. As indicated earlier, without this infection data it is a challenge to correctly interpret the data shown in Figure 4 which is the only experiment in the study that utilized primary cells.
2. If the cell cycle and cytokine measurement data cannot be provided for the infected primary CD4+ T cells, then it is only appropriate for the authors to clarify in their abstract and elsewhere in the text that those findings were only made in the cell line model. Therefore, Lines 29-31 ought to read as follows to avoid giving the incorrect impression to the reader that these findings were also made using primary CD4 T cells: “Our results show a significant enhancement of apoptosis and G2 cell cycle arrest in HUT78 cells infected with R77Q, but not WT NL4-3 or the Vpr Null strain. Conversely, HUT78 cells infected with WT virus show higher levels of necrosis”
In the following sentence of the abstract (Line 32), the authors indicate that the above results were also seen in primary CD4+ T cells. But where is the data to support this statement in the manuscript? This is the reason this reviewer pointed out that the cell cycle and cytokine measurement data in the primary T cells are missing.
3. It had been requested that the authors consider providing quantitative data for the purported multiple Vpr immunoblotting experiments whose representative results are now shown in Figure 2. The authors’ response was that, “We do not have the ability to quantify protein band intensity at this time”. The authors could consider several imaging densitometry software that are available for free or trial download. The quantitative analysis is an important set of data that ought to be included alongside the representative immunoblots.
Author Response
Comments and Suggestions for Authors
- In the renumbered Figure 4 (formerly Figure 5) the authors show that upon infection of cord blood-derived primary CD4+ T cells, the R77Q Vpr mutant virus induces a significantly higher level of apoptosis compared to the WT or null Vpr virus. My reviewer request was NOT for additional experiments, but only to show the extent of infection of these primary T cells with each of these viruses for experiments already conducted and whose data are shown in 5A-C. Similar to the data shown in Figure 1 for infection of the HUT78 cell line, what is the replicative capacity or extent of infection of the WT, R77Q and null virus in the primary T cells? It cannot be assumed that it will be the same as that observed in the cell line model. As indicated earlier, without this infection data it is a challenge to correctly interpret the data shown in Figure 4 which is the only experiment in the study that utilized primary cells.
Response: We agree with this comment, it is likely that the replicative capacity of these viruses is different in a cell line vs in primary cells. The reason why this experiment was initially performed was due to a reviewer request from a previous submission, and the reviewer wanted to know if the apoptotic phenotype would take place in a more relevant cell type, as compared to a cell line. With the 4 different donor samples analyzed to date, we feel very confident that the apoptotic phenotype is confirmed in primary cells, as well as with multiple donors with varying genetics. However, we never quantified viral replication (as in Figure 1) when these primary cell experiments were performed, because the reviewer at the time did not request that. In order to fulfill this current request, we would have to obtain at least three additional cord blood samples, which is very challenging to us at this time. To share with the reader that the lack of this additional work is a noted shortcoming of this manuscript, we have added the following to the Discussion.
“We also recognize that other experiments included here could be repeated in primary CD4+ T cells besides just the induction of apoptosis, which may provide additional insight into viral replication kinetics, cell cycle arrest and pro-inflammatory cytokine production in a more relevant cell model.”
- If the cell cycle and cytokine measurement data cannot be provided for the infected primary CD4+ T cells, then it is only appropriate for the authors to clarify in their abstract and elsewhere in the text that those findings were only made in the cell line model. Therefore, Lines 29-31 ought to read as follows to avoid giving the incorrect impression to the reader that these findings were also made using primary CD4 T cells: “Our results show a significant enhancement of apoptosis and G2 cell cycle arrest in HUT78 cells infected with R77Q, but not WT NL4-3 or the Vpr Null strain. Conversely, HUT78 cells infected with WT virus show higher levels of necrosis”
In the following sentence of the abstract (Line 32), the authors indicate that the above results were also seen in primary CD4+ T cells. But where is the data to support this statement in the manuscript? This is the reason this reviewer pointed out that the cell cycle and cytokine measurement data in the primary T cells are missing.
Response—We appreciate the feedback. After re-reading these sections of the manuscript, we agree that the wording makes implications that are not accurate. We have modified the abstract to be more precise, as follows:
“Acquired immunodeficiency syndrome (AIDS) occurs when HIV depletes CD4+ helper T cells. Some patients develop AIDS slowly or not at all, and are termed Long-Term Non-Progressors (LTNP), and while mutations in the HIV-1 Viral Protein R (vpr) gene such as R77Q are associated with LTNP, mechanisms for this correlation are unclear. This study examines induction of apoptosis, cell cycle arrest, and pro-inflammatory cytokine release in the HUT78 T cell line following infection with replication-competent wild-type strain NL4-3, the R77Q mutant, or a Vpr Null mutant. Our results show a significant enhancement of apoptosis and G2 cell cycle arrest in HUT78 cells infected with R77Q, but not with WT NL4-3 or the Vpr Null strain. Conversely, HUT78 cells infected with WT virus show higher levels of necrosis. We also detected lower TNF and IL-6 release after infection with R77Q vs WT. The apoptotic phenotype was also seen in the CEM cell line and in primary CD4+ T cells. Protein expression of the R77Q Vpr variant was low compared to WT Vpr, but expression levels alone cannot explain these phenotypes because the Null virus did not show apoptosis or G2 arrest. These results suggest that R77Q triggers a non-inflammatory apoptotic pathway which attenuates inflammation, possibly contributing to LTNP. ”
We also modified the Discussion section to be more clear:
“We have noted several phenotypes that differ between the R77Q Vpr mutant and WT Vpr, leading to four main conclusions: 1) the NL4-3-based R77Q Vpr mutation enhances the pro-apoptotic activity of HIV-1 in multiple human T cell lines and in primary human T cells; 2) the NL4-3-based WT and Null viruses kill both CD4+ T cell lines and primary cells via a necrotic, non-apoptotic pathway; 3) the R77Q Vpr mutation significantly enhances G2 cell cycle arrest in HUT78 cells, and 4) the R77Q mutant fails to induce expression of pro-inflammatory cytokines in HUT78 cells to the extent seen by WT virus.”
- It had been requested that the authors consider providing quantitative data for the purported multiple Vpr immunoblotting experiments whose representative results are now shown in Figure 2. The authors’ response was that, “We do not have the ability to quantify protein band intensity at this time”. The authors could consider several imaging densitometry software that are available for free or trial download. The quantitative analysis is an important set of data that ought to be included alongside the representative immunoblots.
Response: We have now added quantification of blots from three separate experiments. The brief summary is that expression of the R77Q isoform is ~6-fold lower than the WT isoform. We appreciate the comment, and this new analysis has strengthened the manuscript. I was under the impression that we would have to go back and re-do the blots so that we could image them from while the signal was still being emitted, but my students researched into how to do the quantification from the images we previously acquired, which we were able to do. The new figure is now included as Figure 2C, and corresponding modifications have been made to Methods, Results, Discussion, Figure Legend 2, and Supplementary Data file.
Methods:
“Quantitative analysis of the chemiluminescent bands on the western blots was done using ImageJ software (version 1.54g, National Institute of Health) and normalized to a GAPDH loading control. The relative quantification of the Vpr band was expressed as a ratio to the uninfected control.”
Results:
“Vpr expression from the R77Q mutant was consistently found at reduced levels compared to the WT virus, across several experiments (see Fig. 2A/2B, and Supplementary Data Set S10). The GAPDH loading control showed that approximately equal amounts of total protein were added for each sample, and the p24 loading control indicated that similar amounts of total HIV-1 proteins were also present in each sample (Fig. 2B). Quantitative analysis of the relative Vpr bands was performed using ImageJ software, and showed that there was a significant ~6-fold reduction in the expression of R77Q compared to WT (p =0.0326) (Fig. 2C).”
Discussion:
“We detected a 6-fold reduction in the detection of Vpr protein in R77Q-infected samples as compared to WT virus by immunoblotting (Fig. 2).”
Figure Legend:
“C) Bar graph indicating the quantitative analysis of the relative western blot bands expressed as a ratio to the uninfected control. Data are representative of 3 independent experiments and error bars indicate SE. Asterisk * indicates that there was a statistically significant difference when compared to all other conditions (p≤0.05) whereas n.s. signifies no statistical significance (p>0.05).”
Round 3
Reviewer 2 Report
Comments and Suggestions for Authors
The authors have made a reasonable effort to address this reviewer's more recent concerns. I have no further comments but will reiterate the as-of-yet unaddressed issue of the authors missing the important data set showing that the primary cells under investigation in Figure 5 are indeed infected by HIV. At this point there is no way to tell that the observations being made in Figure 5 are a consequence of primary cell infection with the HIV Vpr variants (WT, R77Q or null). The authors may want to consider pointing this out to the reader.